# Neural Evidence of Mirror Self-Recognition in the Secondary Somatosensory Cortex of Macaque: Observations from a Single-Cell Recording Experiment and Implications for Consciousness

**DOI:** 10.3390/brainsci11020157

**Published:** 2021-01-25

**Authors:** Rafael Bretas, Miki Taoka, Sayaka Hihara, Axel Cleeremans, Atsushi Iriki

**Affiliations:** 1Laboratory for Symbolic Cognitive Development, RIKEN Center for Biosystems Dynamics Research, Kobe 650-0047, Japan; rafael.bretasvieira@riken.jp (R.B.); miki.taoka@riken.jp (M.T.); 2Program in Brain, Mind & Consciousness, Canadian Institute for Advanced Research, Toronto, ON M5G 1M1, Canada; axcleer@ulb.ac.be; 3Consciousness, Cognition, and Computation Group (CO3), Centre for Research in Cognition and Neurosciences (CRCN), ULB Neuroscience Institute (UNI), Université Libre de Bruxelles (ULB), B-1050 Brussels, Belgium

**Keywords:** self-recognition, consciousness, self-awareness, self-other, self-in-the-world

## Abstract

Despite mirror self-recognition being regarded as a classical indication of self-awareness, little is known about its neural underpinnings. An increasing body of evidence pointing to a role of multimodal somatosensory neurons in self-recognition guided our investigation toward the secondary somatosensory cortex (SII), as we observed single-neuron activity from a macaque monkey sitting in front of a mirror. The monkey was previously habituated to the mirror, successfully acquiring the ability of mirror self-recognition. While the monkey underwent visual and somatosensory stimulation, multimodal visual and somatosensory activity was detected in the SII, with neurons found to respond to stimuli seen through the mirror. Responses were also modulated by self-related or non-self-related stimuli. These observations corroborate that vision is an important aspect of SII activity, with electrophysiological evidence of mirror self-recognition at the neuronal level, even when such an ability is not innate. We also show that the SII may be involved in distinguishing self and non-self. Together, these results point to the involvement of the SII in the establishment of bodily self-consciousness.

## 1. Introduction

Since Gallup’s mirror experiment showed that chimpanzees could recognize their own images reflected in a mirror [1], the mirror test has arguably become a reference for an animal’s ability to develop a conscious sense of self [2]. In its most basic form, mirror self-recognition is manifested by directing behavior toward oneself instead of toward the mirror image. Such behaviors may be elicited by simple conditioned responses, which could be mistaken as self-recognition [3,4]. Nevertheless, this first contingency between one’s own movement and the mirror feedback is an important step in understanding that the agent in the mirror is oneself [5].

Mirror self-recognition has been considered to develop spontaneously in humans and some great apes when habituated to a mirror but not in macaques [6]. However, macaques can acquire this ability after a brief training period [7,8,9], suggesting that the cognitive substrate required for conscious self-awareness is present to some extent. Similarly, children below a certain age have been observed to pass the mirror test when prompted but not spontaneously [10,11]. Despite that, whether different levels of self-awareness can be inferred from acquired versus spontaneous mirror self-recognition is debatable.

While humans can explicitly express through verbal behavior the understanding that the image in the mirror is a representation of the self, other animals lack such ability, making some comparisons flawed [12]. Until now, multiple experiments have shown behavioral evidence of mirror self-recognition in different animals [2,13,14,15], but little is known about its neural underpinnings, which could provide a direct evidence of self-recognition in non-human animals.

In simple terms, neuronal activity in response to directing attention to the self-body is an indication of self-awareness *as one first has to know who one is in order to self-recognize* [16]. In practice, differentiating such an activity from the constant flow of somatic and unconscious sensory information can be a challenge. An ideal candidate target for the study of self-awareness is multimodal somatosensory neurons (i.e., neurons that respond to one or more sensory modalities other than somesthetic stimuli). As somatosensory information is constrained by the limits of the self-body (i.e., cannot be perceived over distance, such as visual or auditory information), multimodal somatosensory neurons represent an ideal target for the study of self-body awareness by tying visual information to tactile and proprioceptive sensations [17]. 

In the primate brain, neurons within the secondary somatosensory cortex (SII) have been found to display responses to both somatosensory [18,19] and visual stimuli [20,21,22]. Despite the tactile responses in this area being topographically organized in somatotopic maps in resemblance to those found in the primary somatosensory cortex (S1) [23,24,25], the maps in the SII are more diffuse and less spatially accurate. Larger body areas appear overrepresented in the SII, sometimes with large bilateral receptive fields (RFs) that may include the whole body [24,26,27], making for a more proportional representation of the physical space rather than of sensory accuracy. The low spatial resolution and specificity of these RFs do not seem to account for the SII’s role in object recognition [28,29] or sensorimotor integration [30,31]. Instead, these results suggest this region’s involvement in whole-limb or whole-body representations [25].

The convergence of somatosensory and visual information in the SII may serve different purposes [32]: increasing perceptual threshold [33], attention [34,35], “cross-cortical traffic hubs” [36], or sensorimotor integration [30]. Another characteristic of SII activity is a distinguished response to actions or stimuli related specifically to the self-body or others’ bodies [22,37]. These functions, coupled with whole-body [26] and multimodal responses [21,36], could be an indication of this region’s key role in merging sensory stimuli in a whole-body schema, which is one of the possible mechanisms in the establishment of self-consciousness [38,39].

To understand whether SII neurons can distinguish the self-image in a mirror, we conducted an exploratory study, observing bilateral single cell activity in an awake Japanese macaque (*Macaca fuscata*). During the experiment, the subject was touched while observing her own body through the mirror. Responses to touching the self, the experimenter, or objects, as well as responses to being touched, were evaluated in cells that displayed visual receptive fields. 

In this article, the SII is defined as the areas englobing both the SII proper and the parietal ventral area. This definition follows the cytoarchitectonic organization and electrophysiological responses found in these regions [40,41,42].

## 2. Materials and Methods

### 2.1. Training and Behavioral Indicators of Mirror Self-Recognition

One female Japanese macaque (*Macaca fuscata*) weighing 8.8 kg was habituated to the experimental conditions for 2 weeks until she gained the ability to recognize her own image in a mirror (Figure 1). The habituation consisted of a visual–proprioceptive association task, during which the animal was gently touched and had her limbs passively moved by the experimenter while she sat in front of a full-body mirror until mirror self-recognition (MSR) was induced. Mirror self-recognition was validated through a variation of the marker test [1,8], in which a laser pointer was shone on a part of the subject’s own body that could only be seen through the mirror. The subject was able to repeatedly touch her own body in the place where the laser shined, indicating that she recognized the body seen in the mirror was her own. To avoid confounding operant-conditioning effects, during neither training nor testing was a reward given in response to any self-directed behavior [7]. However, food was given sparingly for calming and acclimating the subject to the experimental setup. Spontaneous behaviors reflecting MSR were also observed at the end of the training period, such as touch and grooming of body areas that could only be seen through the mirror. All procedures were conducted with the approval of the RIKEN Animal Experimental Committee (permit number H26-2-211 of April 2014) and in strict accordance with the National Institutes of Health Guide for the Care and Use of Laboratory Animals.

### 2.2. Surgery

The subject was anesthetized with sodium pentobarbital anesthesia (30 mg/kg, i.v.), and a custom design headpost made of polyether ether ketone (PEEK) was firmly attached to the cranium with screws made of the same material. A 2-week period was given for recovery, and then head-fixed habituation and training started.

After the training period, a second surgery under similar conditions took place for the implantation of the two recording chambers, one at the right and the other at the left hemispheres. Each chamber was square shaped (28 × 28 mm), covering the entire area of interest (SII proper and PV). The chambers were stereotaxically implanted parallel to the brain surface, perpendicular to the upper bank of the lateral sulcus, having their final positions calibrated through MRI.

### 2.3. Data Acquisition and Recording Sites

After the chamber’s protective cap was removed and the sterile condition of the recording area surface was confirmed, the manipulator (MO-96A, Narishige) was attached to the recording chamber. Varnished tungsten electrodes (1–2 MΩ at 1 kHz, FHC, Inc., Bowdoin, ME, USA) were lowered into the SII for signal acquisition. A recording grid was used to align the electrodes and assure a straight and accurate penetration. The correct positioning of the recording sites and grid was confirmed by visual inspection through the recording chamber of their position over the brain surface and by reconstructing the electrode tracks on MRI.

Neurons were recorded bilaterally on the parietal operculum and neighboring regions, ranging between the lateral and intraparietal sulcus on the longitudinal axis and from 2 mm anterior of the central sulcus until 22 mm posterior to it on the anteroposterior axis (Figure 2). The recording depth extended from the brain surface to a maximum of 10.5 mm to cover the whole depth of the lateral sulcus at the recording site as confirmed via MRI. Neural activity was amplified and monitored in real time through an oscilloscope and loudspeaker. Only when a well-isolated stable single unit was identified would the test procedure and recording start.

### 2.4. Identification of Visual and Somatosensory Responses

Stimulation consisted mainly of the following procedures: the experimenter moving their own hand or body nearby the subject without contact; the experimenter touching the subject; the subject touching the experimenter or an object; or the subject touching her own body. The latter two conditions were carried out passively, with the subject’s arm in a relaxed state being held and moved by the experimenter. If a response to a different form of stimulation (e.g., from the experimenter approaching the subject from behind) was detected during the experiment, the stimuli would be repeated to ascertain any associated activity, with an average of 13 repetitions per stimuli. When required, the subject would have her eyes covered or have the direct view of the stimulation site blocked to evaluate the influence of vision in the neuronal response. The stimulated areas were located in the upper body (from the waist up), including the face and upper limbs. Some of these areas could only be seen through the mirror (e.g., abdomen or dorsum).

Responses were qualitatively assessed in real time by the experimenter during stimulation and always cross-checked by a second experimenter. Twelve responsive neurons were also recorded along with a video of the task and timed to the stimuli’s onset, apex, and end.

## 3. Results

A total of 822 neurons were recorded within the target region in both the right and left hemispheres, with responses to visual stimuli detected in 124 units. Among these visually responsive units, 96 (77%) displayed visual RFs (i.e., responded to stimuli moving within the animal’s peripersonal space or preceding/succeeding touch). The remaining 28 neurons (23%) responded to other complex visual stimuli, such as movement in the vicinity of the animal.

Within the neurons that displayed visual RFs, two categories of complex responses were found: responses to images in the mirror and differential responses to touching the self or non-self with hand. Within the responses to mirror images, self-related vs. non-self-related responses could also be identified (Figure 3). As visual RFs have been previously reported by [20], here, we focus on those other two types of complex responses.

### 3.1. Mirror Image Responses

During the experiment, 36 of the 124 visually responsive neurons were found to respond to stimuli seen through the mirror. These neurons displayed receptive fields that were either located in areas outside of direct view from the animal (30 units, Figure 4A) or were found to respond directly to the mirror image (six units, Figure 4B). These stimuli were further subdivided into self-related when relative to a visual receptive field close to the subject’s body, or non-self-related when in response to general movement behind the animal or to viewing other stimuli in the mirror. Care was taken to ensure that the animal was looking forward toward the mirror during these responses.

### 3.2. Self-Related/Non-Self-Related Touch Responses

Another type of response identified was specific to self or non-self-stimuli during touch with the self-hand. Self-neurons (seven units) responded to the subject touching a specific part of her own body (Figure 4C). Non-self-neurons (five units), on the other hand, responded to touching the experimenter or objects but not when touching the self-body (Figure 4D).

## 4. Discussion

In this study, neurons responsive to visual and somatosensory stimuli were found in the SII, corroborating previous reports on the multimodal nature of this region [21,43,44,45,46]. Furthermore, some cells responded differentially to visual and somatosensory stimuli related to the self versus non-self. Responses were also observed when the stimulated area could only be seen through a mirror, indicating mirror self-recognition at the neuronal level. One non-self-neuron also responded to the experimenter touching their own face in a specific location (Figure 5), which likely represents another sign of self/non-self differentiation in the SII, which will be discussed in Section 4.3.

### 4.1. Study Limitations

As an exploratory study, the present study had significant limitations, which must be addressed taking into consideration the experiment design. The monkey’s hand was moved passively by the experimenter. This was necessary, as the animal would not make such movements spontaneously, with the required number of repetitions, without training. To avoid the controversy of conditioned responses producing behaviors that simulate mirror self-recognition [3,4], a compromise was made with using passive movement instead of training the animal to move on cue. This limitation could account for the lower proportion of neurons found in the touch condition (Figure 3).

Another limitation was using a single subject (albeit recorded bilaterally). Nevertheless, our findings agree with previous observations on the functions of SII in humans and non-human primates [20,21,22,24,25,26,27,28,29,30,31], being conducted following previously tested and published protocols [21,26]. Our goal was to provide a starting point for future studies on the neuronal basis of self-recognition and consciousness.

### 4.2. Single-Cell Activity as An Indicator of Mirror Self-Recognition in the Brain

The relationship between behaviors performed in front of a mirror and self-awareness has been riddled with controversy. Self-recognition-like behaviors could be conditioned or cued [2,3,4,10], and the ability to understand the information from a mirror does not automatically connote awareness of the self-image on such media [47,48]. Therefore, a more direct measure of self-recognition is important.

One solution is to examine whether neuronal activity is associated with different indicators of self-recognition [49]. We found that single neurons in the SII not only responded to the self-image in the mirror but, more importantly, they also exhibited similar responses to viewing the self directly. Such activity occurred despite the radically different points of view—the third-person view in the mirror and the direct first-person view. In addition to visual responses, tactile responses were also observed to be modulated by self and non-self stimuli. Therefore, the activity of these SII neurons could conceivably be used as a direct indicator of self-recognition, which does not require any form of verbal processing [50].

It is important to clarify that these neuronal responses alone do not prescribe *conscious* self-recognition at the subjective level. Nevertheless, the findings provide evidence that the perceptual prerequisites for self-recognition are present in the SII. Rather than being a response to simple sensory information, the neuronal activity in SII may represent a kinesthetic–visual matching of the self-body as viewed in the mirror [51]. The notion that this self-representation reaches conscious awareness at some point still relies, in an absence of verbal communication, on more variations of behavioral indicators such as examining body areas that cannot be directly seen or the marker test. Whether the activity in SII indicates that self-recognition has reached a conscious level requires more investigation. Nonetheless, we speculate that it is at most a single step below awareness, considering the abundant connections of SII to frontoparietal networks involved in conscious perception [22,52,53,54].

### 4.3. Role of SII in Self-Body Consciousness

Previous experiments have reported visual and somatosensory responses of neurons in parietal areas to localized areas or body parts [52,55,56,57]; the neuronal activity in SII differs from those responses, as the SII neurons show enlarged receptive fields, sometimes engulfing the whole body [24,26,27]. Moreover, bimodal responses in the parietal areas strongly prioritize vision over somatosensation, with small and relatively well-defined receptive fields centered toward the head and hand [52,56].

In the present study, the animal was able to see her full body in the mirror, with responses in the SII covering large and multiple areas of the body. In this manner, we could assess responses to the body as a whole and not only to its parts. In fact, *bodily self-consciousness is experienced as the transparent content of a single, coherent whole-body representation rather than as multiple representations of separate body parts* [58]. 

The association between somatosensory and visual information may also be the mechanism through which a sense of self is developed, whereas dissimilarities would form the concept of non-self [53,59]. The same mechanism may be responsible for the emergence of mirror self-recognition as the monkey was habituated to being in front of the mirror. Therefore, the SII would be situated as a fundamental but rudimentary precursor of self-recognition, being one of the first areas in which whole-body somatosensation is integrated [24,26,27], perhaps forming a global self-recognition network [60].

Hihara et al. [21] remarked that complex real-world-like visual stimuli are required to activate the SII, which explains the choice of stimuli in our study. Additionally, by avoiding confounding reward-related factors, the mirror responses could be confidently ascribed to self-recognition over simple conditioned responses [8], reflecting more naturalistic conditions than those found in associative learning experiments. The attenuation or ceasing of somatosensory responses in the absence of visual stimulation (e.g., by covering the eyes), coupled with the maintenance of responses to stimuli that could only be seen through the mirror, confirms that vision is an essential component of self-identification. Visual responses in the SII may originate from connections to frontal and parietal areas [21,45,61,62] that contribute to the control of visually guided forelimb movements and spatial recognition of objects [45,63]. Similarly, information about the sight of touch in others could originate from mirror neurons in areas connected to the SII [22], with coding for others’ body parts arising primarily from self-representation, which also provides a basis for imitation learning and shared interpersonal representations [64]; this hypothesis coincides particularly well with the activity described in Figure 5. However, as the stimuli described in Figure 5 were not part of our standard set, the hypothesis remains largely unexplored. We anticipate that future studies will be able to better assess responses to third-person stimulation in the SII.

Visual attention is another process that may influence the observed results. Being intrinsically involved in conscious awareness, attention has been shown to modulate SII activity and somatosensory perception [34,35]. We confirmed during the experiment that the animal had a clear sight of the stimulated site, with changes in the firing rate being consistent among trials. Contrary to being a confounding factor, the influence of attention through vision suggests that the SII is a candidate region for the synchronization of fronto-parietal attention networks [65,66] and temporo-parieto-occipital phenomenal consciousness networks [67]. This would support the network synchronization theory (NetSync) [68], which accounts for the parallel interaction between multiple brain areas in the formation of conscious experience [69,70,71]. It remains to be tested whether any response specific to the self in the mirror would be detected in the SII before mirror habituation, which is an indication of an underlying evolutionary role of having a sense of self [72].

While parietal neurons seem to code a reaching space centered on the head [52,56,57], and hippocampal cells code a navigation space based on distal cues and reward locations [73,74,75], the SII appears to situate the whole body in the surrounding environment [60]. As the SII codes not only the separate body parts as a source of somatosensation but also a combination of those body parts into an indivisible concept of an individually and socially aware self-body, both mirror recognition and self/non-self differentiation could be explained along the observed sensory responses.

## 5. Conclusions

Our findings provide neural evidence of mirror self-recognition in the primate SII, which was corroborated by coincident self-related and non-self-related activity, supporting this area’s involvement in the establishment of bodily self-consciousness [38]. Moreover, the differentiation between the experimenter and objects at the neural level also indicates a broader role of the SII in attributing consciousness to other beings. The location of the SII in the intraparietal sulcus, with connections to posterior parietal regions involved in the conscious experience of external stimuli as well as to proprioceptive primary areas [45,66,76], places the SII in an ideal position to process a body-in-the-world map [53], integrating sensory information from the external world and the self. 

## Figures and Tables

**Figure 1 brainsci-11-00157-f001:**
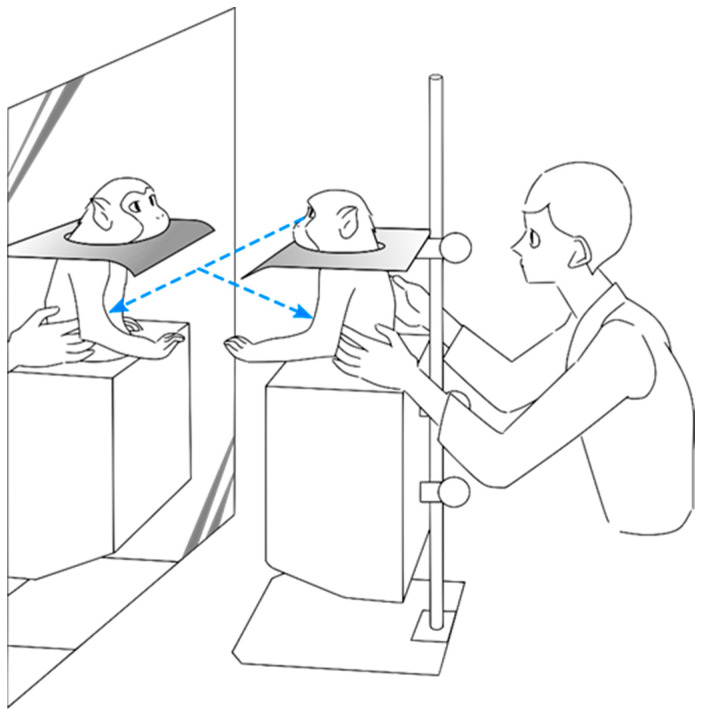
Experimental setup used during the experiment. The animal sat in a monkey chair that neither covered the upper body nor restricted upper limb movement. A full-body mirror was placed in front of the animal. When needed, an opaque plate could be placed to ensure that the animal was not able to view her own body directly, but only through the mirror reflection.

**Figure 2 brainsci-11-00157-f002:**
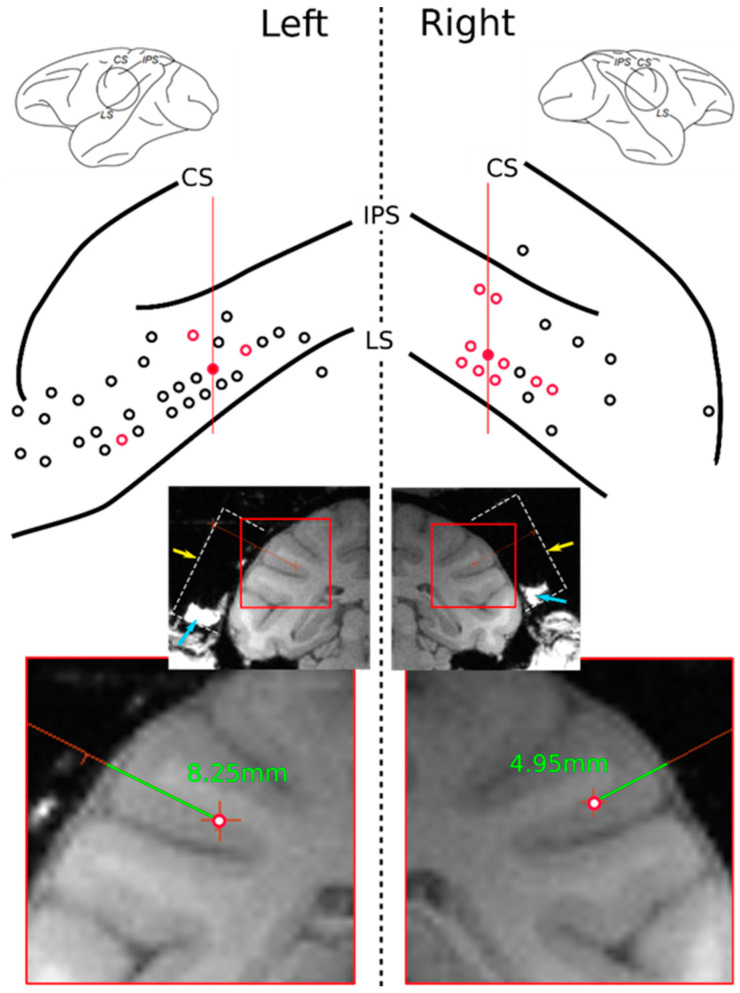
Upper: Electrode penetration sites (red and black circles) on the surfaces of the left and right hemispheres. Red circles indicate locations where self-related mirror responsive neurons were found. Central, intraparietal, and lateral sulci indicated as CS, IPS, and LS, respectively. Red lines indicate the position of the MRI slices shown below. Lower: Coronal MRI slices of the recorded subject show the recording chamber’s location (dashed white line, reconstructed). Yellow arrow points to the top of the chamber where the recoding grid was positioned. Blue arrows indicate tissue and liquid naturally accumulated within the bottom walls of the chamber. A cropped detail below shows reconstructed electrode tracks (green lines) of two examples of mirror responsive neurons found in the red-filled tracks on the upper drawing with their respective depths of recording.

**Figure 3 brainsci-11-00157-f003:**
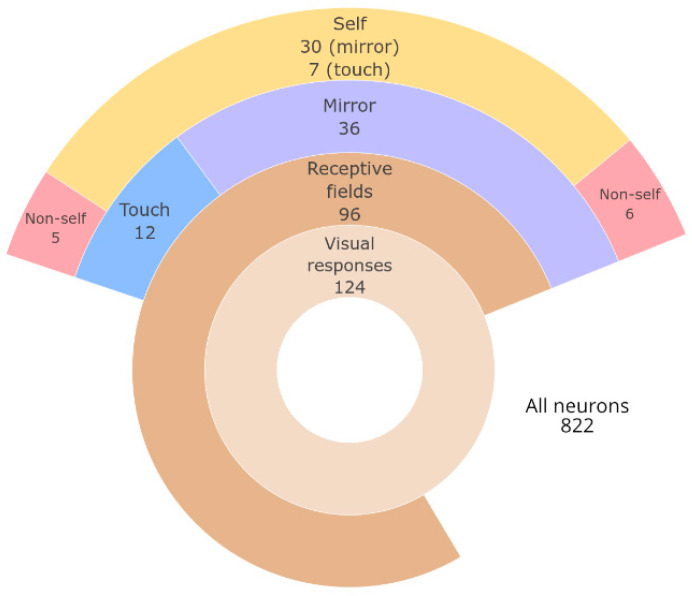
Summary of neurons found within each category of response. Size of ring sectors drawn in proportion to the number of neurons.

**Figure 4 brainsci-11-00157-f004:**
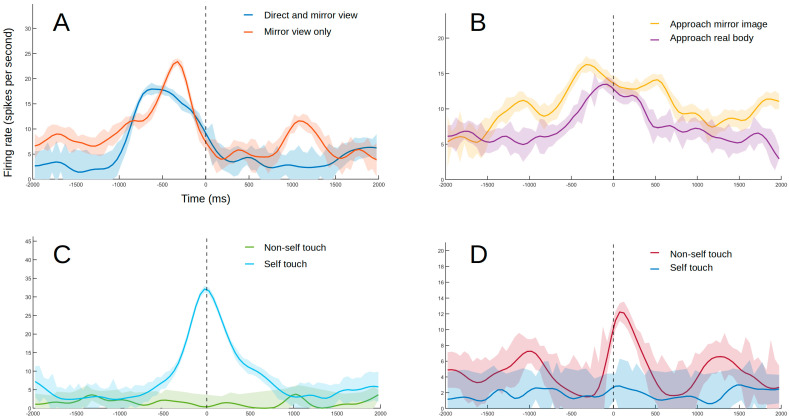
Example of responses from different types of neurons. The lines indicate smoothed average firing rates for all trials (Gaussian-weighted moving average of ten 50 ms samples). The shaded areas indicate the coefficient of variation (not smoothed) as a measure of consistency in firing rate between trials. (**A**) Example of a neuron with a visual receptive field near the self-body with responses maintained when seen through the mirror. The experimenter moves their hand toward the monkey’s abdomen without touching it. The results were confirmed after blocking the subject’s peripheral view with an opaque plate, with the stimulation site becoming visible only through the mirror. The dashed line indicates the moment of closest approach. (**B**) Example of a neuron responding to direct stimulation to the image in the mirror. The experimenter’s hand approaches the subject’s face projection in the mirror. The response to the hand approaching the subject’s real face follows a similar trend. The dashed line indicates the moment of closest approach. (**C**) Example of self-touch neuron. The monkey’s arm is moved by the experimenter until the hand touches the self-mouth. No response was observed when the same movement was repeated with an object sandwiched between the animal’s mouth and hand (non-self touch condition). No response was detected when mouth was touched by experimenter or external object either. The dashed line indicates touch onset. (**D**) Example of non-self touch neuron. The monkey’s left hand is touching the experimenter’s hand. No response was observed when touching the subject’s own opposite hand or forearm while replicating the same movement that was made when touching the experimenter’s hand (blue line). The dashed line indicates touch onset.

**Figure 5 brainsci-11-00157-f005:**
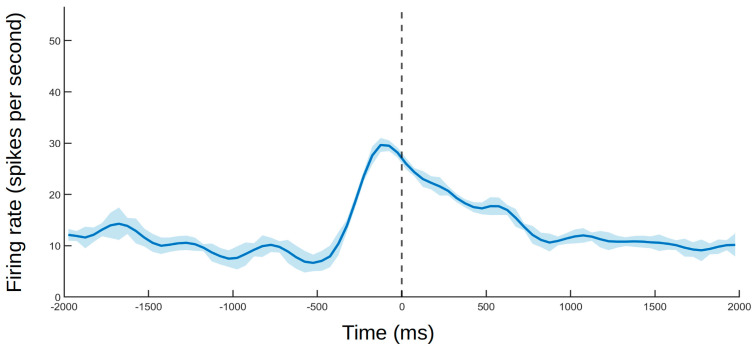
Example of a neuron that responded to seeing the experimenter touching their own cheek. The dashed line indicates touch onset. The solid line and shaded area follow the same standard as that in Figure 4.

## Data Availability

The data presented in this study are available on request from the corresponding author. The data are not publicly available due to containing information that could compromise the privacy of the researchers involved.

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
