# Peer review of "Neural Evidence of Mirror Self-Recognition in the Secondary Somatosensory Cortex of Macaque: Observations from a Single-Cell Recording Experiment and Implications for Consciousness"

_brainsci, 2021, doi:10.3390/brainsci11020157_

Round 1

Reviewer 1 Report

In this study, Rafael and colleagues investigated the neural evidence of mirror self-recognition in the secondary somatosensory cortex of the monkey. The authors claim that the monkey acquired the ability of mirror self-recognition after training. During the experiment, the monkey underwent visual and somatosensory stimulation, multimodal visual and somatosensory activity was detected in the SII, with neurons found to respond to stimuli seen through the mirror. SII neurons also could distinguish self-related stimulus from non-self-related stimulus. The deals is interesting, but there may be some gap between the conclusion about SII - self-consciousness and the results.

There were also several issues about the experimental design, in particular extra variables. First, four experiment conditions were included in this design, and experimenter held and moved subject’s arm in the latter two condition. However, this causes a confusion between self-touch and non-self touch, experimenter touch the subject simultaneously when subject performed touching tasks. Besides, no more details about the number of repetitions and duration time were mentioned.

For the results in Figure 3A-B about the mirror view, there was only one condition where the monkey can see their own body. If there were any chance that the neurons showed the same response when the monkey saw another monkey or a monkey picture which replace the view in the mirror? Please clarify.

Moreover, the author claimed that visual attention could influence the observed results. However, it does not appear to be explicitly mentioned about the control of subject’s attention. How do the authors control for that?

The result in Figure 4 is not very convincing. Because only one non-self-neuron showed this response, but there were 5 neurons responded to non-self touch. I think more evidence is needed in this regard.

Why use covariance as the shaded area in the figure instead of standard error? Please give more detailed information in the method part.

Author Response

In this study, Rafael and colleagues investigated the neural evidence of mirror self-recognition in the secondary somatosensory cortex of the monkey. The authors claim that the monkey acquired the ability of mirror self-recognition after training. During the experiment, the monkey underwent visual and somatosensory stimulation, multimodal visual and somatosensory activity was detected in the SII, with neurons found to respond to stimuli seen through the mirror. SII neurons also could distinguish self-related stimulus from non-self-related stimulus. The deals is interesting, but there may be some gap between the conclusion about SII - self-consciousness and the results.

We are very grateful for the reviewer’s comments and for their time reading and providing great constructive comments to our manuscript. We appreciate the understanding of the limitations on the type of study proposed. We hope we could satisfactorily address the reviewer’s concerns and improve the quality of our work through the changes introduced below, following the guidance received.

There were also several issues about the experimental design, in particular extra variables. First, four experiment conditions were included in this design, and experimenter held and moved subject’s arm in the latter two condition. However, this causes a confusion between self-touch and non-self touch, experimenter touch the subject simultaneously when subject performed touching tasks. Besides, no more details about the number of repetitions and duration time were mentioned.

As an exploratory experiment, we tried to make the experiment as broad as practically possible. To clarify the limitations in this study, we have included a new section on the discussion: “4.1. Study limitations” (from line 214). The type of experiment has also been disclosed in the introduction:

Line 73: “To understand whether SII neurons can distinguish the self-image in a mirror, we conducted an exploratory study, observing bilateral single cell activity in an awake Japanese macaque (Macaca fuscata).”

Line 225: Our goal was to provide a starting point for future studies on the neuronal basis of self-recognition and consciousness.”

About the animal arm being moved by the experimenter, unfortunately that was a compromise required to avoid reward-based training. This animal, as it seems to be a recurrent characteristic of Japanese Macaque, did not produce enough spontaneous behaviors of the type to allow a clear effect in the neural activity. We have included more details about this limitation in the new “4.1. Study limitations” section, as well as the average number of repetitions in the methods:

Line 216: “The monkey’s hand was moved passively by the experimenter. This was necessary as the animal would not make such movements spontaneously, with the required number of repetitions, without training. To avoid the controversy of conditioned responses producing behaviors that simulate mirror self-recognition [3,4], a compromise was made with using passive movement instead of training the animal to move on cue. This limitation could account for the lower proportion of neurons found in the touch condition (Fig. 3).”

Line 144: “If a response to a different form of stimulation (e.g., from the experimenter approaching the subject from behind) was detected during the experiment, the stimuli would be repeated to ascertain any associated activity, with an average of 13 repetitions per stimuli.”

For the results in Figure 3A-B about the mirror view, there was only one condition where the monkey can see their own body. If there were any chance that the neurons showed the same response when the monkey saw another monkey or a monkey picture which replace the view in the mirror? Please clarify.

Indeed, there was no control in the form of another monkey or a picture/video as it is common in human subject experiments (reference [12] discusses the topic as well as some of the limitations involved). By showing responses to both seeing stimuli to the self body directly as well as through the mirror, we assume the monkey understands the mirror-image as the self (and not another monkey). We have also shown the touch responses differentiating the self and the experimenter. We have updated the text with some points related to the issue raised by the reviewer as well as pointing that as a limitation that our study does not address:

Line 280: Similarly, information about the sight of touch in others could originate from mirror neurons in areas connected to the SII [22], with coding for others' body parts arising primarily from self-representation, which also provides a basis for imitation learning and shared interpersonal representations [64] – a hypothesis that coincides particularly well with the activity described in Fig. 5. However, as the stimuli described in Fig. 5 were not part of our standard set, the hypothesis remains largely unexplored. We anticipate that future studies will be able to better assess responses to third-person stimulation in the SII.”

Moreover, the author claimed that visual attention could influence the observed results. However, it does not appear to be explicitly mentioned about the control of subject’s attention. How do the authors control for that?

 As we did not intend to investigate the effects of attention, we simply confirmed the animal could see the stimuli (either directly or through the mirror). We have clarified it in the discussion when talking about attention:

Line 289: We confirmed during the experiment that the animal had a clear sight of the stimulated site, with changes in the firing rate being consistent among trials.

In fact this entire paragraph (from line 268) was rewritten, with some extra relevant discussion added.

The result in Figure 4 is not very convincing. Because only one non-self-neuron showed this response, but there were 5 neurons responded to non-self touch. I think more evidence is needed in this regard.

The stimuli reported on Figure 5 unfortunately was not among our repertoire of consistently tested stimuli. Despite that, we found such a response interesting, with the possibility of being connected to the broader theme of consciousness, therefore we included it as a mere example of a possible response that could be deeper explored in future studies.

Following the reviewer’s criticism, we made clearer that it was an isolated finding, not part of our standard set of stimuli:

Line 280: Similarly, information about the sight of touch in others could originate from mirror neurons in areas connected to the SII [22], with coding for others' body parts arising primarily from self-representation, which also provides a basis for imitation learning and shared interpersonal representations [64] – a hypothesis that coincides particularly well with the activity described in Fig. 5. However, as the stimuli described in Fig. 5 were not part of our standard set, the hypothesis remains largely unexplored. We anticipate that future studies will be able to better assess responses to third-person stimulation in the SII.

Why use covariance as the shaded area in the figure instead of standard error? Please give more detailed information in the method part.

That was a writing mistake on our part and we apologize. We meant coefficient of variation, not covariance. The text has been corrected, thank you very much for pointing it out.

We have also checked the standard error of the mean for the data, but found it to be too forgiving of random noise as well as outliers that could possibly skew the data. We choose the coefficient of variation as a more robust indication of the relative dispersion in the firing rate between trials, showing that the responses were indeed consistent among all trials and not only a fluke caused by isolated outliers. We updated the figure description explaining what we hope to indicate with such measure:

Line 183: “The lines indicate smoothed average firing rates for all trials (Gaussian-weighted moving average of ten 50ms samples). The shaded areas indicate the coefficient of variation (not smoothed) as a measure of consistency in firing rate between trials. “

Reviewer 2 Report

The ‘major’ comments here cover the fact that this study has limitations and these should be addressed in the Intro/Disc. This is a nice study. That being said, what it ISN’T is important to address and frame. It is not a multiple N study, it is not comparative, it is not a pre/post design, it is limited to certain brain areas. To deal with these, I suggest additions to the Intro/Disc

This may be a unique point as the major issue starts in the first paragraph. I would like to see the first paragraph expanded to 2 or 3 paragraphs. I understand the briefness of the flow of the Intro. However, the many critics of this paper will want a more detailed account of why spontaneous is so critical. Since this paper ultimately does not test chimps, orangutans, or humans, the Intro and Discussion must talk in more detail about training vs. spontaneous recognition. The alternative to this is actually running this exact study as a comparative study which is clearly a different study. I think the authors would do well to argue against themselves here, that is, be skeptical that a) trained SR is the same as spontaneous SR and b) think that trained SR indicates self-awareness. Since many reading this will at a minimum skeptical, better the authors frame the argument.

The second major point in the paper is also an Intro/Discussion issue. There are data (our lab has seen it so I know it’s real) that SMA, MC, and Somatosensory areas are involved in SR and many aspects of ToM. That being said, compared to TPJ, MPFC, lateralized PFC and even precuneous, motor areas are not what one thinks when one thinks self-awareness. Thus, there needs to be a big(ger) discussion here about what the brain areas are and is it just ‘action’ SR or are the authors suggesting that if I remove the SII that SR would disappear. Again, we can derive studies (TMS the animals) that would test this directly but that is not this paper. Rather than collect more data, I would suggest a stronger literature discussion here. Honestly, readers will want to know what the authors think here-how CRITICAL is this are to SR or self-awareness?

Lastly. The authors don’t do a pre-post design. That is, record before and after training. This too limits what one knows and this should be addressed. Would these neurons be dormant pre-training?

Personally, I would like an image on the surgery/recording sites. Also video if possible.

Author Response

Reviewer 2

The ‘major’ comments here cover the fact that this study has limitations and these should be addressed in the Intro/Disc. This is a nice study. That being said, what it ISN’T is important to address and frame. It is not a multiple N study, it is not comparative, it is not a pre/post design, it is limited to certain brain areas. To deal with these, I suggest additions to the Intro/Disc.

We are very grateful for the reviewer’s comments and for their time reading and providing constructive comments to our manuscript. We appreciate the understanding of the limitations on the type of study proposed. We hope to satisfactorily address the reviewer’s concerns and improve the quality of our work through the changes introduced below following the guidance received.

Indeed, our experiment has important limitations. One of the main reasons is the lack of studies on the SII, of which functions we are still trying to pinpoint. At the reviewer’s suggestion, we have clarified the exploratory nature of the study on the introduction:

Line 73: “To understand whether SII neurons can distinguish the self-image in a mirror, we conducted an exploratory study, observing bilateral single cell activity in an awake Japanese macaque (Macaca fuscata).”

We have also prepared a new section on the discussion “4.1. Study limitations” that better talks about those issues.

Line 214: “4.1. Study limitations

As an exploratory study, the present study had significant limitations, which must be addressed taking into consideration the experiment design. The monkey’s hand was moved passively by the experimenter. This was necessary as the animal would not make such movements spontaneously, with the required number of repetitions, without training. To avoid the controversy of conditioned responses producing behaviors that simulate mirror self-recognition [3,4], a compromise was made with using passive movement instead of training the animal to move on cue. This limitation could account for the lower proportion of neurons found in the touch condition (Fig. 3).

Another limitation was using a single subject (albeit recorded bilaterally). Nevertheless, our findings agree with previous observations on the functions of SII in humans and non-human primates [20-22,24-31], being conducted following previously tested and published protocols [21, 26]. Our goal was to provide a starting point for future studies on the neuronal basis of self-recognition and consciousness.”

This may be a unique point as the major issue starts in the first paragraph. I would like to see the first paragraph expanded to 2 or 3 paragraphs. I understand the briefness of the flow of the Intro. However, the many critics of this paper will want a more detailed account of why spontaneous is so critical. Since this paper ultimately does not test chimps, orangutans, or humans, the Intro and Discussion must talk in more detail about training vs. spontaneous recognition. The alternative to this is actually running this exact study as a comparative study which is clearly a different study. I think the authors would do well to argue against themselves here, that is, be skeptical that a) trained SR is the same as spontaneous SR and b) think that trained SR indicates self-awareness. Since many reading this will at a minimum skeptical, better the authors frame the argument.

We extended the first paragraph into 3 paragraphs, introducing more details on the mirror test, it’s limitations, some discussion on spontaneous vs. trained and cued behavior:

Line 30: “Since Gallup's mirror experiment showed that chimpanzees could recognize their own images reflected in a mirror [1], the mirror test has arguably become a reference for an animal's ability to develop a conscious sense of self [2]. Mirror self-recognition is, in its most basic form, manifested by directing behavior towards oneself instead of towards the mirror image. Such behaviors may be elicited by simple conditioned responses, which could be mistaken as self-recognition [3,4]. Nevertheless, this first contingency between one’s own movement and the mirror feedback is an important step in understanding that the agent in the mirror is oneself [5].

Mirror self-recognition has been considered to develop spontaneously in humans and some great apes when habituated to a mirror but not in macaques [6]. However, macaques can acquire this ability after a brief training period [7-9], suggesting that the cognitive substrate required for conscious self-awareness is present to some extent. Similarly, children below a certain age have been observed to pass the mirror test when prompted but not spontaneously [10,11].

While humans can explicitly express through verbal behavior the understanding that the image in the mirror is a representation of the self, other animals lack such ability, making some comparisons flawed [12]. Until now, multiple experiments have shown behavioral evidence of mirror self-recognition in different animals [2,13–15], but little is known about its neural underpinnings, which could provide a direct evidence of self-recognition in non-human animals.”

The discussion has also been rewritten, with a section on how direct neuronal activity can be used as an indicator of mirror self-recognition in the brain (line 227: section “4.2. Single-cell activity as an indicator of mirror self-recognition in the brain”)

The second major point in the paper is also an Intro/Discussion issue. There are data (our lab has seen it so I know it’s real) that SMA, MC, and Somatosensory areas are involved in SR and many aspects of ToM. That being said, compared to TPJ, MPFC, lateralized PFC and even precuneous, motor areas are not what one thinks when one thinks self-awareness. Thus, there needs to be a big(ger) discussion here about what the brain areas are and is it just ‘action’ SR or are the authors suggesting that if I remove the SII that SR would disappear. Again, we can derive studies (TMS the animals) that would test this directly but that is not this paper. Rather than collect more data, I would suggest a stronger literature discussion here. Honestly, readers will want to know what the authors think here-how CRITICAL is this are to SR or self-awareness?

We suggest the SII to have a role as a precursor of SR (and therefore mirror SR), maybe as one of the first areas where somatosensory information is integrated in a whole-body shape, and not only it’s separate parts (as references 24-26 indicate). Unfortunately the lack of more data on the SII functions specifically related to SR does not allow us to extrapolate much beyond the reported findings. Still, in humans, SII lesions do not seem to critically impair self-recognition, therefore it could be argued that the functions performed by the SII are fundamental but rudimentary enough to be performed by other areas as well (in humans).

We have clarified the idea of the SII as a precursor for SR on the discussion, which was rewritten and separated in 3 parts, one of which is “4.2. Single-cell activity as an indicator of mirror self-recognition in the brain”:

Line 227: “4.2. Single-cell activity as an indicator of mirror self-recognition in the brain

The relationship between behaviors performed in front of a mirror and self-awareness has been riddled with controversy. Self-recognition-like behaviors could be conditioned or cued [2] 3,4,10], and the ability to understand the information from a mirror does not automatically connote awareness of the self-image on such media [47,48]. Therefore, a more direct measure of self-recognition is important.

One solution is to examine whether neuronal activity is associated with different indicators of self-recognition [49]. We found that single neurons in the SII not only responded to the self-image in the mirror but, more importantly, also exhibited similar responses to viewing the self directly. Such activity occurred despite the radically different points of view – the third-person view in the mirror and the direct first-person view. In addition to visual responses, tactile responses were also observed to be modulated by self and non-self stimuli. Therefore, the activity of these SII neurons could conceivably be used as a direct indicator of self-recognition, which does not require any form of verbal processing [50].

It is important to clarify that these neuronal responses alone do not prescribe conscious self-recognition at the subjective level. Nevertheless, the findings provide evidence that the perceptual prerequisites for self-recognition are present in the SII. Rather than being a response to simple sensory information, the neuronal activity in SII may represent a kinesthetic-visual matching of the self-body as viewed in the mirror [51]. The notion that this self-representation reaches conscious awareness at some point still relies, in absence of verbal communication, on more variations of behavioral indicators such as examining body areas that cannot be directly seen or the marker test. Whether the activity in SII indicates that self-recognition has reached a conscious level requires more investigation. Nonetheless, we speculate that it is at most a single step below awareness considering the abundant connections of SII to frontoparietal networks involved in conscious perception [22,52-54].”

Lastly. The authors don’t do a pre-post design. That is, record before and after training. This too limits what one knows and this should be addressed. Would these neurons be dormant pre-training?

Those neurons are still somatosensory neurons, coding for receptive fields (both visual, as well as somatosensory), therefore we expect that, without mirror self-recognition, most of this activity would not be present or weaker in response to the mirror image. The rewritten discussion, besides including the section on the limitations of the study and the nature of the neural activity in response to mirror images (as previously mentioned), also clarified that we unfortunately don’t have a clear answer to that question:

Line: 295: “It remains to be tested whether any response specific to the self in the mirror would be detected in the SII before mirror habituation, an indication of an underlying evolutionary role of having a sense of self [72].”

Personally, I would like an image on the surgery/recording sites. Also video if possible.

At the reviewer’s request, we have included a figure with the recording sites (fig. 2) and MRI images from the subject tested.

As this is an open-access publication, it may not be appropriate to publish pictures or videos of the animal visible to the public, although we are open to show the videos and pictures of the implanted chamber personally under request.

Round 2

Reviewer 2 Report

This paper is much improved by the changes. The authors really put work into the revisions and it shows. This is a much more measured, reserved paper, and that actually makes the claims more important. For example, the Figure is incredible and it really helps with both the science, but also the general interest of the paper. This is really nice work. I would add but a few remarks. 

Mirror self-recognition has been considered to develop spontaneously in humans and some great apes when habituated to a mirror but not in macaques [6]. However, macaques can acquire this ability after a brief training period [7-9], suggesting that the cognitive substrate required for conscious self-awareness is present to some extent. Similarly, children below a certain age have been observed to pass the mirror test when prompted but not spontaneously [10,11].

1) I would add a sentence at the end stating that we really don't know what this indicates for the broader 'self-awareness' question. That is, are animals more self-aware when they are trained? Are animals that don't need training more self-aware? I think "Further research is needed to parse out the differences in self-awareness between those that self-recognize spontaneously and those that do not"

Author Response

We would like to thank the reviewer once more for the comments and we are glad the revision was satisfactory.

About the issue raised on the second round of review: we have added a sentence at the end of the paragraph (as suggested) composed taking as basis the reviewer's suggestion and comments:

Mirror self-recognition has been considered to develop spontaneously in humans and some great apes when habituated to a mirror but not in macaques [6]. However, macaques can acquire this ability after a brief training period [7-9], suggesting that the cognitive substrate required for conscious self-awareness is present to some extent. Similarly, children below a certain age have been observed to pass the mirror test when prompted but not spontaneously [10,11]. Despite that, whether different levels of self-awareness can be inferred from acquired versus spontaneous mirror self-recognition is debatable.